# Multiscale red blood cell hitchhiking for targeted deep tissue gene delivery in lungs

Kyung Soo Park[1,2], Vineeth Chandran Suja [1,2], Jayoung Kim[1,2,7], Danika Rodrigues [1,2], Malini Mukherji[1,2], Maithili Joshi [1,2], Yongsheng Gao[1,2,8], Michael Griffith Bibbey [1,2], Jeong-Won Choi[1,2], Rick Liao [1,2], Morgan E. Janes[1,2,3], Metecan Erdi [1], Andrés Da Silva Candal[4], David L. Cameron[5,6], Julian A.N.M. Halmai[5,6], Kyle D. Fink [5,6], Samir Mitragotri [1,2] ✉ & Bijay Singh [1,2] ✉

The clinical impact of gene therapies is constrained by poor delivery to target tissues beyond the liver after intravenous administration. Current molecular targeting strategies, such as capsid engineering or gene-carrier surface modification, have achieved only limited success due to their inability to overcome the hierarchical barriers from injection to deep tissue transduction. Here, we introduce a Multiscale Approach using RBC-mediated hitchhiking and Vascular Endothelium Leakage (MARVEL), which integrates red blood cell hitchhiking with VEGF-induced vascular permeabilization to enhance accumulation and penetration of cargoes. Using adeno-associated viruses (AAVs) as a model, MARVEL markedly increases AAV localization in the lungs, improves endothelial transcytosis, and enables gene expression in deeper tissue layers while maintaining a favorable safety profile. We further demonstrate that MARVEL can be adopted into an in situ hitchhiking approach, bypassing the need for ex vivo formulation. MARVEL provides a scalable strategy to address long-standing delivery challenges in gene therapy.

Adeno-associated viruses (AAVs) have become a leading gene delivery platform for multiple indications owing to their high in vivo transduction efficiency and safety profile in animal and human[1,2]. Attempts have been made to improve the efficacy of AAV-based therapeutics by engineering tissue tropism, increasing transduction efficiency, and decreasing immunogenicity[3–5]. These improvements have led to promising preclinical and clinical results in recent years, leading to the FDA approval of AAV-based therapeutics for Duchenne muscular dystrophy and Leber congenital amaurosis, among others[6–9].

Despite their clinical success, the broad applicability of AAV-based gene therapy is limited by the challenges associated with targeting specific cells and tissues outside the liver upon intravenous administration. The challenge lies in the fact that AAVs need to overcome multiple hierarchical hurdles after intravenous administration before exhibiting tissue and cell-specific delivery, including (i) avoiding the liver clearance by accumulating on the endothelium of the target tissue, (ii) crossing the endothelium to infiltrate the target tissue, and (iii) transducing the target cells within the tissue.

[1]John A. Paulson School of Engineering and Applied Sciences, Harvard University, Cambridge, MA, USA. [2]Wyss Institute of Biologically Inspired Engineering, Harvard University, Boston, MA, USA. [3]Harvard-MIT Division of Health Sciences and Technology, Massachusetts Institute of Technology, Cambridge, MA, USA. [4]Clinical Neurosciences Research Laboratory, Clinical University Hospital, Health Research Institute of Santiago de Compostela, Santiago de Compostela, Spain. [5]Neurology Department, Stem Cell Program and Gene Therapy Center, UC Davis Health System, Sacramento, CA, USA. [6]MIND Institute, UC Davis Health System, Sacramento, CA, USA. [7]Present address: Department of Pharmaceutical Sciences, College of Pharmacy, University of North Texas Health, Fort Worth, TX, USA. [8]Present address: Department of Bioengineering, The University of Texas at Dallas, Richardson, TX, USA. ✉e-mail: mitragotri@seas.harvard.edu; bijaysingh@seas.harvard.edu

Past efforts to improve AAV-mediated gene therapy have focused primarily on virology, aiming at serotype selection and modification of the transgene cassette and capsid components[10–15]. However, without explicitly addressing the delivery challenges, the applications of AAVs remain restricted to cases where delivery hurdles are less relevant, for example, directly injecting AAVs into the tissue, such as the retina and muscle, or exploring applications in the liver, the natural target site of AAVs after intravenous administration[16–18]. Some efforts have aimed at engineering the AAV to improve tissue tropism. However, simultaneous improvement of tissue accumulation and transduction in deeper tissue regions has proved challenging.

The challenge in achieving high tissue tropism can be appreciated from the *master targeting* Eq. (1) (detailed description in Supplementary Information 1 and Supplementary Fig. 1), which describes the flux, $J$, of the capsid, or any other gene carrier, into tissues after injection into a blood vessel.

$$J = C_o \lambda_{tis} P_{vas} e^{-\frac{t}{\tau}} \qquad (1)$$

where $C_o$ is the initial carrier concentration in blood upon injection, $\lambda_{tis}$ is the enhancement of local tissue concentration of the carrier achieved by the targeting strategy, $P_{vas}$ is the vascular endothelial permeability in the target tissue, and $\tau$ is the time constant of blood clearance of the carrier. Classical targeting dogma uses polyethylene glycol (PEG) to increase $\tau$, and relies either on targeting ligands to increase $\lambda_{tis}$ or enhanced permeation and retention (EPR) to achieve high $P_{vas}$ for tumor targeting. The efficacy of such approaches is limited by the inherent chemical conflict between PEG-mediated enhanced circulation and ligand-mediated improved vascular binding. The efficacy of ligand binding is further limited by inadequate margination of cargoes to the vascular endothelium. This lowers the likelihood that the targeting ligands actually engage with their receptors on vascular endothelium. The variability of EPR also makes it a challenge to rely entirely on it for targeting purposes.

Our previous studies have shown that drug loading on red blood cells (RBC), called RBC hitchhiking, increases the accumulation of nanoparticles in various organs, including the lungs, brain, and kidneys[19–25]. Protein loading on RBC using polyphenols, such as tannic acid, has been demonstrated since the 1950s for various applications[26,27]. Using a similar approach, we have previously loaded AAVs onto the RBC surface to target-deliver AAVs to the lungs and reduce their off-target gene expression[28]. While the increased AAV concentration in tissue endothelium enabled by RBC hitchhiking offers an increased likelihood of deeper tissue penetration, the hitchhiking strategy is not inherently designed to overcome the endothelial barrier.

Here, we address that challenge by incorporating the vascular endothelial growth factor (VEGF) into the carrier design. Several prior studies have shown that VEGF improves endothelial permeability through the VEGF receptor-2 (VEGFR-2) signaling pathway, which triggers several downstream pathways related to vascular permeabilization[29]. Leveraging this ability of VEGF, we report a Multiscale Approach using RBC-mediated hitchhiking and Vascular Endothelium Leakage (MARVEL), which leverages (i) RBC-mediated improved circulation ($\tau$), (ii) RBC-mediated enhanced tissue accumulation ($\lambda_{tis}$), and (iii) VEGF-mediated vascular permeabilization ($P_{vas}$). MARVEL uses concurrent adsorption of AAVs and VEGF on the RBC surface using polyphenol-mediated complexation. Upon intravenous administration, RBC-adsorbed AAVs and VEGF are dislodged through contact- and shear-mediated transfer within the vascular endothelium, which occurs on a microscale. This increases local AAV concentration and VEGF-mediated vascular permeabilization, leading to submicroscale deep tissue penetration of AAVs (Fig. 1). Furthermore, we demonstrate that hitchhiking can be achieved in situ, where AAV-VEGF complexes, upon intravenous injection, bind to native RBCs and achieve organ-targeted, deep-tissue transduction. This simplified process demonstrates MARVEL's potential for translation and applicability to various indications, thus offering a tool for effective gene therapy.

## Results
### RBC formulation preparation
RBCs were loaded with AAV alone (RBC/AAV) or AAV + VEGF (MARVEL) by complexation with tannic acid (TA) and iron chloride (III) ($FeCl_3$) to induce metal-phenolic network formation (Fig. 2A). TA and $FeCl_3$ concentrations were first optimized with AAV alone to minimize RBC hemolysis and aggregation during formulation (Supplementary Fig. 2). The AAV, TA, and $FeCl_3$ complexes were visible on the surface of RBCs after formulation, in contrast to the smooth surfaces of bare RBCs (Fig. 2B). Based on the observed modification on the surface, we investigated the rheology of TA and Fe-coated RBCs by measuring their viscosity (Supplementary Fig. 3). The viscosity of coated RBCs did not show a statistical difference compared to bare RBCs at physiological shear rates ($>40 \, s^{-1}$).

### Functional characterization of RBC/AAV complex
AAV loading can be controlled by adjusting the concentration of AAV in the incubation solution (Fig. 3A). The loading efficiencies were similar between AAV serotypes 6 and 9, demonstrating potential orthogonality to various tropism-engineered AAVs of the MARVEL (Supplementary Fig. 4). AAVs are stably associated with RBCs and released in a shear-dependent manner. When RBC/AAV was placed at

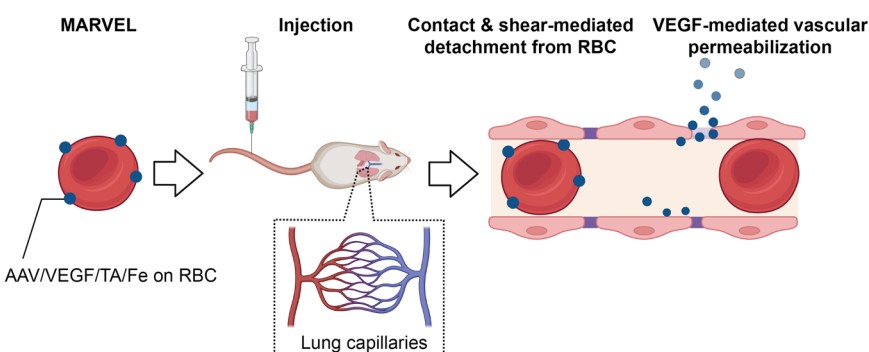

**MARVEL** **Injection** **Contact & shear-mediated detachment from RBC** **VEGF-mediated vascular permeabilization**

AAV/VEGF/TA/Fe on RBC

Lung capillaries

**Fig. 1 | Schematic description of RBC hitchhiking.** RBCs with surface-loaded AAV and VEGF are administered via intravenous or intra-arterial routes, after which they enter peripheral vessels in target organs (shown in the schematic: lung capillaries after intravenous injection). As the RBC complex squeezes through the capillaries, AAV and VEGF are dislodged through contact and shear forces. VEGF permeabilizes the endothelium by disrupting cellular junctions, allowing AAV infiltration and transduction in deeper tissues. TA: tannic acid. Created in BioRender. Park, C. (2025) https://BioRender.com/i75h519.

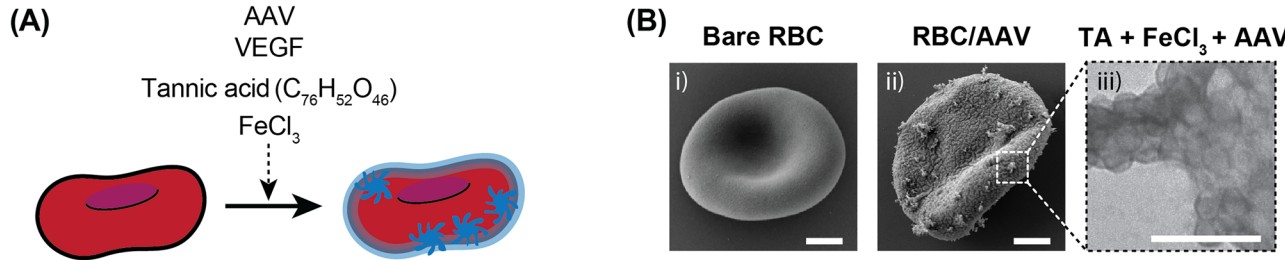

**Fig. 2 | Formulation process of RBC hitchhiking. A** Schematic description of MARVEL preparation through one-pot synthesis. **B** SEM images of i) bare RBC, ii) RBC loaded with AAV, and iii) TA/Fe/AAV complex. Representative images from two independent experiments. Scale bars (B) i) and ii) = 1 μm, and iii) = 100 nm.

room temperature with mild shaking (250 rpm), a cumulative % AAV release of 40% was observed within 48 h (Fig. 3B). In contrast, when the RBC/AAV was exposed to shear stresses by placing under flow conditions, more than 65% of the AAV was released within 10 min, reaching nearly 95% release at 0.1 Pa in 2 min (Fig. 3C), demonstrating shear-induced release of AAVs from the RBC surface.

The ability of AAVs loaded on RBC to transfer to endothelial cells was confirmed ex vivo in a microfluidic chip laden with endothelial cells (EA.hy926) on the channel surface (Supplementary Fig. 5). Significantly higher cell-associated AAVs were observed in the RBC/AAV group than in the free AAV group (Fig. 3D, red signal indicates Alexa Fluor 647-labeled AAVs). Quantitatively, RBC association led to a 3-fold increase in AAV deposition onto the endothelial cells (Fig. 3E). Even under static conditions, RBC-associated AAVs were able to induce gene expression, but the expression was significantly improved under the flow, supporting the role of the shear-induced detachment of AAVs from RBCs (Fig. 3F). The degree of transduction under the flow condition was significantly higher for RBC/AAV than for free AAVs (Supplementary Fig. 6), indicating the significance of the combination of physical contact provided by RBC hitchhiking and shear-induced release of AAVs to maximize transduction.

RBC hitchhiking led to a significantly higher accumulation of AAVs in the lungs in vivo after tail vein injection compared to free AAVs (Fig. 3G). While free AAVs accumulated primarily in the liver, the RBC/AAVs accumulated mainly in the lungs, resulting in a 90-fold higher lung-to-liver ratio for the RBC/AAV than the free AAVs (Fig. 3H, I). When used with GFP-encoded AAVs, RBC/AAVs successfully transduced cells throughout the lungs as assessed 3 weeks post-injection (Supplementary Fig. 7A). The difference in expression between free AAVs and RBC/AAVs was also evident 12 weeks post-injection (Supplementary Fig. 7B). The transgene expression by RBC hitchhiking in off-targets, including the spleen and liver, was comparable to free AAV, and did not amplify the immune response based on the total IgG titer levels in serum (Supplementary Figs. 8, 9). The target tissue for expression can be altered by changing the injection location, thus changing the location for contact-mediated detachment of AAVs from RBCs. For example, when RBC/AAV was injected via the carotid artery, transgene expression in the brain significantly increased compared to free AAVs (Supplementary Fig. 10).

### VEGF induces junctional breach for deep-tissue transduction
VEGF was loaded onto RBCs by co-incubating AAVs and VEGF with TA, FeCl₃, and RBC through one-pot synthesis to accomplish vascular endothelial permeabilization and deep tissue expression. The VEGF loading on RBCs can be controlled by adjusting the feed amount (Fig. 4A). VEGF did not adversely impact AAV loading on RBCs (Supplementary Fig. 11). We also confirmed the loading of a model protein, FITC-BSA, which exhibits a negative charge at neutral pH, in contrast to VEGF, which has a positive charge, demonstrating MARVEL's flexibility in loading cargoes of different therapeutic interests (Supplementary Fig. 12). VEGF maintained its bioactivity after complexation with TA and Fe (VEGF/TA/Fe) as indicated by its ability to impact vascular

endothelial cadherin (VE-cadherin) signals in endothelial cells (Fig. 4B). Specifically, upon incubation with endothelial cells, VEGF released from the VEGF/TA/Fe complex disrupted the junctional proteins, decreasing the VE-cadherin signal comparable to the positive control (Fig. 4B). The cell surface area, perimeter, and Feret's diameter were significantly increased in VEGF/TA/Fe-treated group, supporting junctional disruption (Supplementary Fig. 13). In a transwell setting, free AAVs or RBC/AAVs with or without VEGF were applied to a layer of EA.hy926 cells (Supplementary Fig. 14). Only upon loading VEGF on RBCs (MARVEL) was a statistically significant increase in AAV translocation compared to other groups, likely due to the increased AAV and VEGF concentrations near the endothelial cells.

VEGF, either in the free form or loaded on MARVEL, was administered by a tail vein injection in vivo (Fig. 4C). While free VEGF, which was co-delivered with free AAV, did not lead to a detectable increase in lung VEGF concentration, MARVEL led to significant lung accumulation of VEGF. MARVEL also led to increased lung transgene expression compared to RBC/AAV. Four weeks after the injection, MARVEL showed a trend of higher endothelial transduction than RBC/AAV (Fig. 4D) and a significant increase in epithelial transduction (Fig. 4E). Notably, RBC/AAV without VEGF did not effectively transduce the epithelial cells. However, MARVEL significantly increased the transduction, attesting to the significance of VEGF in inducing deep tissue transfection.

MARVEL was well tolerated in all animals. Four hours after injection, neither AAV + VEGF combination nor MARVEL showed an increase in IL-6 and TNF-α levels in the lungs, while the systemic levels of IL-6 and TNF-α returned to baseline within an hour after injection (Supplementary Fig. 15A, B). AAV or RBC/AAV, with or without VEGF, did not cause mouse bodyweight change, indicating the tolerance of the formulation at the tested doses (Supplementary Fig. 15C).

### In situ drug loading on RBC in whole blood allows lung-targeted delivery
We next assessed whether MARVEL allows AAVs to bind to and hitchhike on RBCs in situ spontaneously, since bypassing ex vivo attachment could substantially simplify the administration process.

Given that the association of MARVEL-AAVs ex vivo with RBCs upon incubation is rapid and that RBCs are the predominant cell type in blood, we hypothesized that MARVEL-AAVs may also associate with RBCs upon direct intravenous injection. We first validated this possibility using a model drug carrier, fluorescence-labeled polystyrene (PS) beads of 200 nm in diameter. PS beads were added to whole blood with or without TA and Fe. The binding between the PS beads and RBC was significantly greater when combined with TA and Fe (Fig. 5A). To assess how serum proteins affect PS bead binding on RBC, we evaluated binding efficiency across varying serum concentrations, with and without TA and Fe. Binding efficiency decreased notably at 10% serum and above, suggesting that serum proteins compete with PS beads during binding (Supplementary Fig. 16A). However, TA and Fe (MARVEL) markedly enhanced PS bead binding to RBCs under all conditions, consistent with Fig. 5A. Notably, the fold increase in

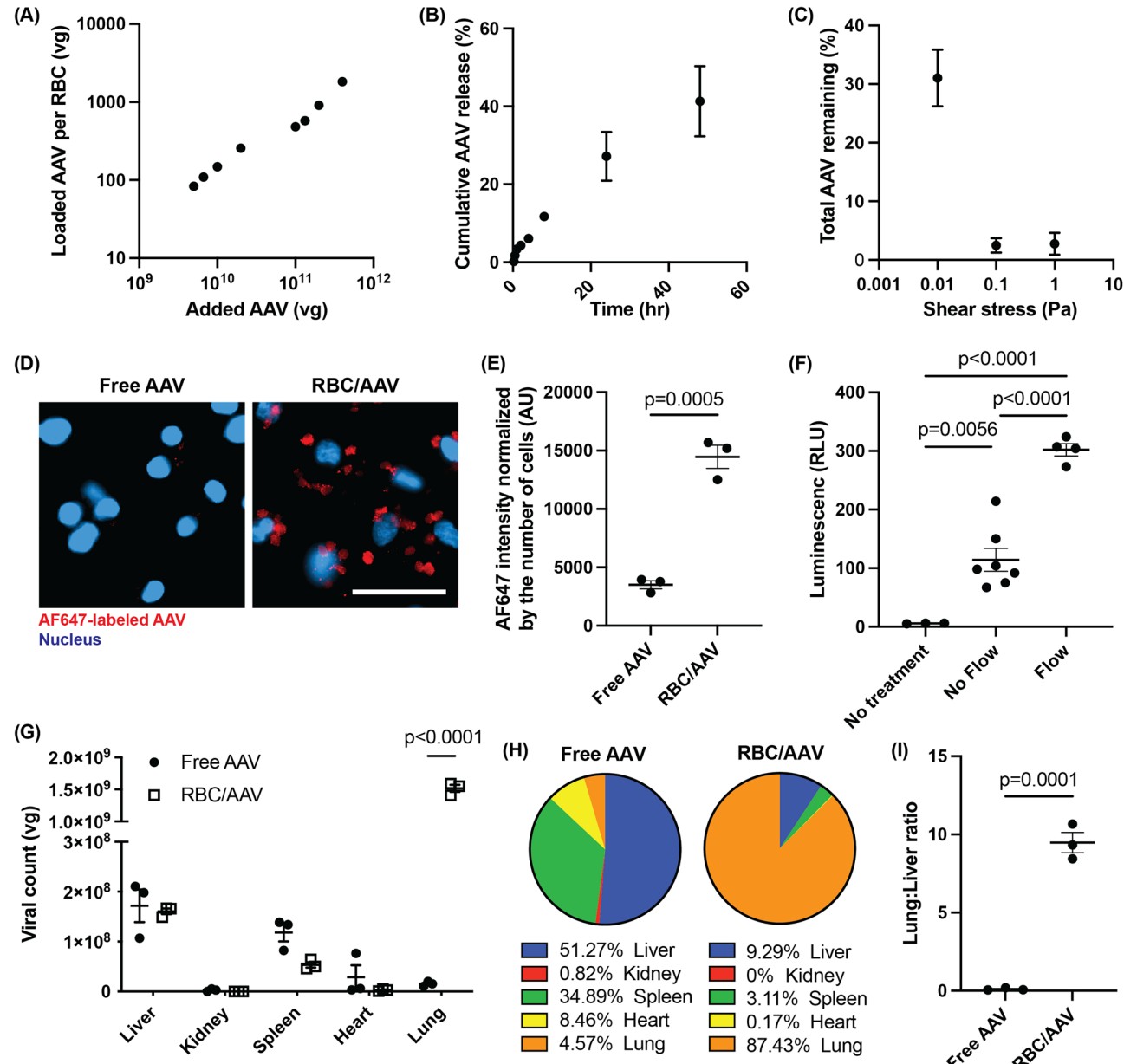

**Fig. 3 | RBC/AAV complex preparation and function characterizations in vitro and in vivo. A** The number of AAV6-CMV-Luc loaded per RBC ($n = 3$ independent biological replicates), **B** cumulative release in a mild shaking condition ($n = 4$ independent biological replicates), and (**C**) shear-induced release were measured by PCR ($n = 3$ independent biological replicates). **D** Confocal images of EA.hy926 cultured in a microfluidic chip were taken after a stream of AF647-labeled AAV6-CMV-Luc was given. AF647 and the nucleus are shown in red and blue, respectively. Representative images from two independent experiments. Scale bar = 50 μm. **E** AF647 signals were quantified from 3 independent planes on the confocal images. **F** In vitro transduction of EA.hy926 cells by RBC/AAV (AAV6-CMV-Luc) treated in a

static or flow condition ($n = 3$ biological replicates for No Treatment, $n = 7$ for No Flow, $n = 4$ for Flow). Consolidated data from two independent experiments. **G** In vivo biodistribution of AAV6-CMV-Luc 1 h after intravenous injection measured with PCR, (**H**) percent distribution among the major organs, and (**I**) lung-to-liver ratio ($n = 3$ female C57BL/6 mice per group, 5–6 weeks of age). Statistical analysis was performed with unpaired, two-sided Student's *t*-test (**E, I**), two-way ANOVA, followed by Sidak's multiple comparisons test (**G**), and one-way ANOVA, followed by Tukey's multiple comparisons test (**F**). All data are presented as mean ± SEM. Source data are provided as a Source Data file.

binding mediated by MARVEL rose with increasing serum concentration, which plateaued at 30% serum concentration (Supplementary Fig. 16B). RBC association of PS beads in the presence of TA and Fe also occurred under flow conditions (Supplementary Figs. 17A, B). PS beads successfully adhered to RBCs within a short time (<2 s) in the presence of TA and Fe. Real-time imaging of the whole blood mixed with PS beads, TA, and Fe during flow cytometry confirmed the in situ binding of PS beads to RBC (Supplementary Fig. 18). A similar study was performed with lipid nanoparticles (LNPs) containing a model mRNA encoding luciferase. The addition of TA and Fe significantly improved

the binding of LNPs to RBC in whole blood, demonstrating the potential generalizability for drug loading onto RBCs within the blood in situ (Supplementary Fig. 19).

AAV formed complexes in the presence of TA and Fe, and these complexes bound to RBCs in whole blood (Fig. 5B, Supplementary Fig. 20A, B). Upon direct intravenous injection, MARVEL-AAV led to the accumulation of AAVs in the lungs within an hour, giving a trend of improved lung-to-liver ratio (based on absolute titers) compared to AAV injection alone (Fig. 5C, D). Notably, the AAV accumulation per mg of organ increased by 6.2-fold in the lungs and decreased by 2.3-fold

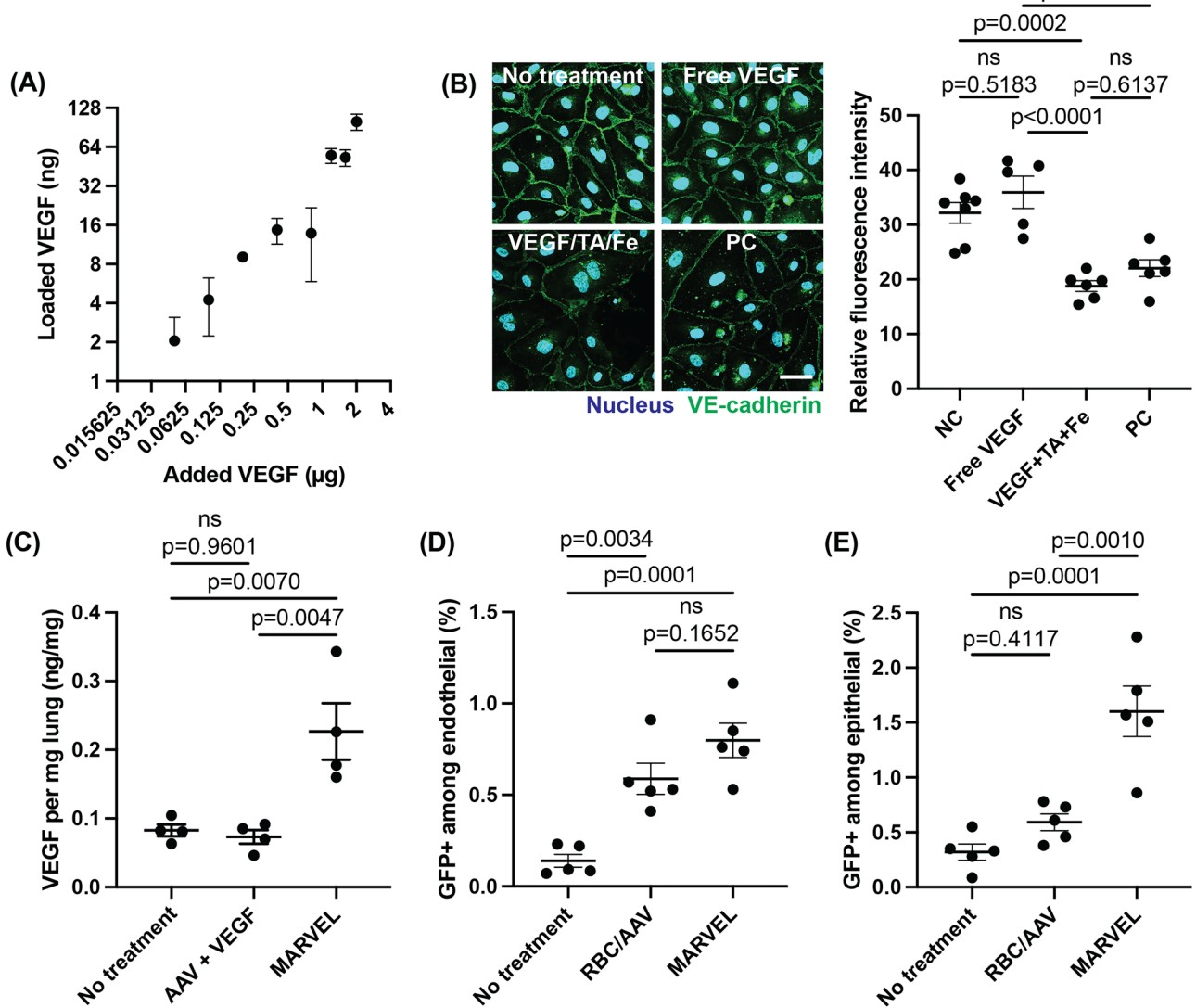

**Fig. 4 | MARVEL preparation and functional characterization in vitro and in vivo. A** VEGF loading on RBC was measured with ELISA ($n = 3$ independent biological replicates). **B** Confocal images of primary human brain endothelial cells after a 30-min incubation with 20 ng/mL VEGF with or without TA and Fe. No treatment (NC) served as the negative control, and a 24-h incubation with 100 ng/mL VEGF was used as the positive control (PC). Representative images are shown from $n = 3$ biological replicates per group. Blue: nucleus; green: VE-cadherin. Scale bar = 50 μm. VE-cadherin signals quantified from $n = 7$ planes (NC), $n = 5$ (free VEGF), and $n = 6$ (VEGF + TA+Fe and PC) from the confocal images. **C** VEGF amounts in the lungs 4 h after intravenous injection measured by ELISA ($n = 4$ female C57BL/6 mice per group, 5–6 weeks of age). The percentages of transduced (**D**) endothelial and (**E**) epithelial cells characterized by GFP+ signals in the lungs were measured with flow cytometry 30 days after intravenous injection with formulations containing AAV6-CMV-GFP and VEGF ($n = 5$ female C57BL/6 mice per group, 5–6 weeks of age). Statistical analysis was performed with (**C, B, D, E**) one-way ANOVA, followed by Tukey's multiple comparisons. Data are presented as mean ± SEM. ns: not significant. Source data are provided as a Source Data file.

and 2.7-fold in the liver and spleen, respectively, with MARVEL-AAV. Thirty days after injection, the mice injected with MARVEL-AAV exhibited significantly greater transgene expression in the endothelial (Fig. 5E) and epithelial (Fig. 5F) cells compared to AAV alone. MARVEL-AAV led to a 4-fold improvement of endothelial gene expression and a 2.6-fold improvement of epithelial gene expression compared to free AAV, which were 388- and 52-fold compared to the respective no-treatment baseline signals. These results confirm that MARVEL improves gene expression through a combination of RBC hitchhiking-mediated lung accumulation of AAVs on the endothelium and subsequent deeper tissue transduction into the epithelial tissue achieved by VEGF.

MARVEL did not increase transgene expression in the liver compared to AAV alone, suggesting minimal off-target transduction and toxicity of the approach (Supplementary Fig. 21). Moreover, in situ

MARVEL was well tolerated by the animals based on the histological assessment by a pathologist on the major organs harvested 30 days after intravenous injection. No differences were observed between the organs of treated and control animals (Supplementary Figs. 22A, 22B). Lastly, analyses of blood chemistry and hematology on blood and serum samples collected 1 and 7 days after intravenous injection indicated that MARVEL was well-tolerated by animals (Supplementary Fig. 23).

## Discussion

In this study, we introduced and validated MARVEL as a platform strategy for the targeted delivery of AAVs and enabling their deep-tissue transduction. Traditional drug delivery strategies, which often focus on molecular-level targeting modifications, such as PEGylation or targeting ligand conjugation, can pose inherent constraints since

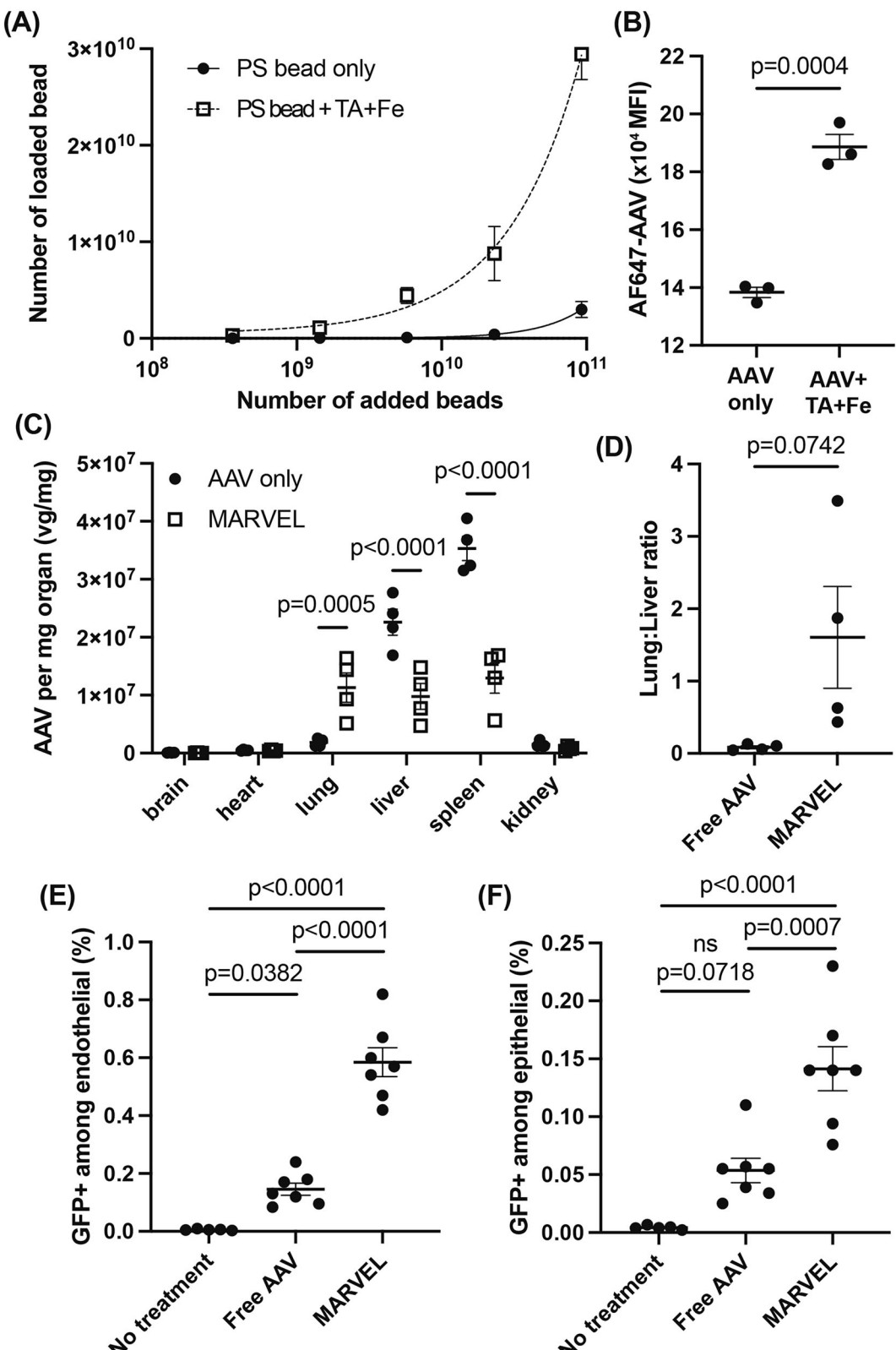

they apply competing chemistries at the same molecular scale, potentially limiting their combined effectiveness. Further, the efficacy of molecular-scale targeting ligands is limited by the low likelihood of these ligands making molecular-scale contact with their corresponding receptors. MARVEL employs a multi-scale approach to propose a dogma in drug delivery: achieving organ-targeted drug accumulation at the microscale and enhancing drug infiltration into deeper tissue

regions at the sub-microscale. By combining the RBC hitchhiking approach with VEGF-mediated vascular permeabilization, we achieved a significantly improved accumulation of AAVs in lung tissues through a concurrent improvement in $\lambda_{tis}$, $P_{vas}$, and $\tau$, surpassing the efficacy of traditional methods based on free AAV (detailed description in Supplementary Information 2 and Supplementary Fig. 24). This dual-action approach increased the concentration of AAVs at the endothelial

**Fig. 5 | In situ drug loading on RBC in whole blood and lung-specific targeting.**
**A** Loading of fluorescent polystyrene beads (PS) with or without TA and Fe on RBC in whole blood in a static condition, measured with a plate reader ($n = 3$ independent biological replicates). **B** Binding of AAV6-CMV-Luc to RBCs in whole blood under static conditions confirmed by flow cytometry ($n = 3$ independent biological replicates). **C** Biodistribution of AAV6-CMV-Luc 1 h after intravenous injection of AAV alone or mixed with TA and Fe into the tail vein, and (**D**) the corresponding lung-to-liver ratio ($n = 4$ female C57BL/6 mice per group, 5–6 weeks of age). Percentages of transduced (**E**) endothelial and (**F**) epithelial cells characterized by GFP

+ signals in the lungs measured by flow cytometry 30 days after intravenous injection of formulations containing AAV6-CMV-GFP (female C57BL/6 mice per group, 5–6 weeks of age; $n = 5$ for the no treatment group, and $n = 7$ for the treatment groups). Statistical analysis was performed with (**B**, **D**) unpaired, two-sided Student's $t$-tests, (**C**) two-way ANOVA, followed by Sidak's multiple comparisons test, or (**E**, **F**) one-way ANOVA, followed by Tukey's multiple comparisons test. Data are presented as mean ± SEM. ns: not significant. Source data are provided as a Source Data file.

interface and facilitated their transcytosis into deeper tissue layers, leading to enhanced gene expression in target cells. The enhanced lung transduction with MARVEL is especially significant, given that AAV6 used in our lung studies has already been optimized for lung delivery by engineering tropism[30]. This suggests that MARVEL can be broadly applied to enhance the targeted transduction of recombinant AAVs (rAAV) orthogonal to existing engineering approaches for specific tropisms. Polyphenols interact with surrounding molecules through non-covalent interactions, including hydrogen bonding, hydrophobic and electrostatic interactions, and van der Waals forces, which gives them versatility in loading payloads with diverse physicochemical properties[31]. The loading of other model therapeutic cargoes, including proteins, polymers, and LNPs, onto the RBC surface in whole blood further underscores MARVEL's versatility, potentially broadening its application to other therapeutic agents and disease targets. We found the LNP loading results particularly promising, as they suggest an approach to enhance genetic transfection via non-viral vectors—a rapidly advancing area in the field with growing applications for LNP technologies[32,33].

Our assessment of histology, blood chemistry, hematology, serum cytokines, and IgG levels indicates that MARVEL does not exacerbate AAV-induced toxicity or immunogenicity. Nevertheless, use of AAV vectors as therapeutic cargos should be taken with caution, since AAV therapies frequently trigger immune responses, raising concerns about potentially severe toxicities and limiting the feasibility of re-dosing strategies. Prior work by Zhao et al. demonstrated that RBC-mediated delivery may enable repeat dosing[28]. Furthermore, the enhanced and sustained transduction achieved via RBC hitchhiking may reduce the need for frequent dosing or enable effective single-dose regimens, which could be advantageous in clinical settings. Finally, the applicability of RBC hitchhiking has been demonstrated in species beyond mice using model drugs, supporting its translational potential for AAV-mediated gene delivery[20]. As an alternative, RBC extracellular vesicles may be an option for nucleic acid delivery for gene therapy[34].

Overall, MARVEL represents a significant advancement in gene therapy, offering a robust and scalable method for overcoming the long-standing challenges of targeted delivery and tissue-specific transduction. The multi-scale delivery approach is a fundamentally different approach and a paradigm shift from traditional targeting methods. It offers an orthogonal strategy to achieve spatially and quantitatively improved drug delivery to target tissues in vivo. Future studies will optimize this system for other target tissues and expand its utility in treating a broader range of genetic disorders.

## Methods
### Ethics statement and animals
All animal experiments were conducted in accordance with protocols approved by the Institutional Animal Care and Use Committee (IACUC) of the Faculty of Arts and Sciences, Harvard University (protocol number 18-02-320-1). Female C57BL/6 mice (5–6 weeks of age) were purchased from the Jackson Laboratory (Bar Harbor, ME). For experiments involving carotid arterial injections, mice 10 weeks of age were used. All studies were performed using female mice. Mice were housed

under a 12-h light/12-h dark cycle at $22 \pm 2\,°C$ and 40–60% relative humidity.

### RBC preparation
Blood was collected from mice via submandibular bleeding using 4 mm lancets and placed in EDTA-coated blood collection tubes. The blood was centrifuged at 1000 g for 10 min at 4 °C. The serum and the buffy coat were removed by pipetting. The RBC pellet at the bottom was transferred into a 15 mL tube filled with ~14 mL of ice-cold PBS. The sample was centrifuged at 650 g for 15 min at 4 °C. The supernatant was removed, followed by resuspension of the pellet with 15 mL of cold PBS. The washing step was repeated twice. After removing the supernatant following the final centrifugation, the packed RBC at the bottom of the tube was transferred into a new tube using a pipette, and 9x the volume of cold PBS was added to achieve a 10% hematocrit RBC solution.

### Ex vivo MARVEL preparation
The stock AAV (AAV6-CMV-GFP and AAV6-CMV-Luc; SignaGen Laboratories) solution containing 5e + 11 AAV and 10 μg of recombinant murine VEGF-165 (#450-32; Peprotech) was mixed with PBS to a 390 μL intermediate volume. 10 μL of 10 mg/mL TA (Sigma-Aldrich, 16201) (dissolved in deionized water) was added, followed by a brief vortex to mix. The sample was incubated at room temperature for 2 min. Next, 100 μL of RBC (10% hematocrit) in PBS was added. The solution was mixed manually by inverting the tube 3–4 times. 3 μL of 10 mg/mL Iron (III) chloride hexahydrate ($FeCl_3$; Sigma-Aldrich, 236489) (dissolved in deionized water) was added, followed by an addition of 500 μL of PBS. The tube was manually inverted several times for mixing, then centrifuged at 100 g for 4 min. The supernatant was removed, and the pellet was resuspended with 1 mL of PBS. This washing step was repeated twice. After the final centrifugation, the pellet was resuspended with 90 μL of PBS to a final volume of 100 μL of RBC/AAV formulation.

For in situ MARVEL, 1e + 11 vg AAV, 450 ng VEGF, 20 μg TA, and 6 μg Fe were mixed in water and adjusted to 1X PBS by mixing 10X PBS to a final volume of 200 μL.

### Microfluidic shear stress-induced AAV release from RBC
Microfluidics chips (Ibidi uChips 0.2 Luer) were treated with 100 μL of poly-L-lysine for 20 min at RT, then washed with 100 μL PBS twice. 100 μL of RBC/AAV (1.25e + 9/mL) was treated on each chip and incubated for 20 min on ice. The chips were washed three times with cold PBS to remove any unbound RBC/AAV. Next, the flow was turned on for 2 min using a peristaltic pump in a closed-loop system with cold PBS in the tubing. The entire chip's channels were imaged using a bright-field microscope to count the RBC/AAV before and after the flow.

### Transduction study in microfluidic chips
To prepare microfluidics chips coated with a model endothelial cell line, EA.hy926 (CRL-2922; ATCC), cells were seeded in microfluidics chips (Ibidi uChips 0.2 Luer) in 100 μL of 3e + 6 cells/mL concentration in complete DMEM media (supplemented with 10% FBS and 1% Pen-Strep), and incubated at 37 °C for 3 h to allow the cells to adhere. The chips were washed twice with 120 μL of fresh media to remove any unbound cells and were further incubated for two days prior to the AAV6-CMV-Luc treatment, with daily media changes. Depending on

## Table 1 | Thermal cycle

| Step 1 | 95 °C 5 min |
|---|---|
| Step 2 | 95 °C 15 s |
| Step 3 | 55 °C 1 min |
| Repeat steps 2-3 39 times | |
| Step 4 | 4 °C Done |

## Table 2 | List of primers and probes

| **CMV primers and probe** | |
|---|---|
| CMV_FP Sequence | TCA TAT GCC AAG TAC GCC CC |
| CMV_RP Sequence | CCC GTG AGT CAA ACC GCT AT |
| CMV_probe Sequence | /56-FAM/TG GGA CTT T/ZEN/C CTA CTT GGC AGT AC |
| **EGFP primers and probe** | |
| EGFP_FP Sequence | AGC AAA GAC CCC AAC GAG AA |
| EGFP_RP Sequence | GGC GGC GGT CAC GAA |
| EGFP_FAM_probe Sequence | /56-FAM/CG CGA TCA C/ZEN/A TGG TCC TGC TGG |

the experimental design, RBC/AAV or free AAV were treated either by a syringe, pipette, or tubing. After treatment, chips were thoroughly washed with complete DMEM and further incubated for 10–14 days until analysis.

### In vitro functional study using endothelial cells

Primary human brain endothelial cells (Cat. ACBRI376; pediatric male donor; Cell Systems) were cultured according to the manufacturer's instructions. For the functional assay, cells were seeded into an 8-well confocal chamber at a density of 3e + 5 cells per well and maintained in an EGM-2 medium (Lonza). After two days, once a monolayer had formed, cells were cultured with cAMP, hydrocortisone, and other supplements to enhance junctional function. On the day of the experiment, cells were treated with 20 ng/mL of free VEGF or 20 ng/mL of VEGF/TA/Fe. After a 30-min incubation, the medium was replaced with fresh medium, and the cells were cultured for an additional 6 h. For the positive control, cells were treated with 100 ng/mL of VEGF and incubated for 24 h. Following the designated incubation times, cells were fixed, blocked, and immunostained with an anti-VE-cadherin antibody (SantaCruz, sc-9989). Samples were washed three times before incubation with a secondary antibody (Invitrogen, A11001) and counterstained with DAPI. Confocal images were captured using a Zeiss LSM980 with Airyscan. Fluorescent intensity of the junctional marker and cellular morphology were quantified using ImageJ.

### Fluorophore labeling on AAV

The NHS-amine reaction chemistry was used to label AAV with Alexa Fluor 647. 1 mg of lyophilized Alexa Fluor™ 647 NHS Ester (succinimidyl ester) (Invitrogen; A20006) was dissolved in 135 μL of 20 mM phosphate buffer containing 1e + 12 vg AAV-CMV-Luc. The solution was incubated at room temperature for an hour under constant shaking. The labeled AAVs were diluted by adding 715 μL of PBS and then purified by running through 7k Zeba desalting columns (Thermo Scientific; 89891) twice. The resulting solution was concentrated using a 3k Amicon Ultra-0.5. The resulting Alexa Fluor 647-labeled AAV was quantified using PCR following the protocol described later in the Methods section.

### Transmission electron microscopy (TEM) imaging

Samples were prepared at a high concentration (>1 mg/mL). Hexagonal carbon TEM grids were placed on a glass cover slide wrapped in parafilm, followed by glow discharge cleaning (Pelco Easyglow) on the grids. Next, on the surface of a stretched piece of parafilm taut across a glass bench surface, 15 μL drops of each sample were placed. Inverted carbon grids were placed on top of the drops (carbon side down) and waited 1 min. The grids were retrieved with fine-tipped forceps, handling only the grid border. The carbon surface was washed with 5 drops of 3% uranyl acetate in deionized water, passed through a 0.22 μm syringe filter. Then, the grid edges were blotted to remove excess moisture, followed by air drying overnight. The resulting samples were imaged on a Hitachi 7800 TEM.

### Scanning electron microscopy (SEM) imaging

Samples were fixed with a solution containing 2% glutaraldehyde overnight at 4 °C and adsorbed onto poly-L-lysine coated 12 mm² coverslips. Samples were dehydrated by incubating in increasing concentrations (50%, 75%, 90%, 95%, 100%, 100%, 100%) of ethanol for 10 min, then dried using a critical point dryer. Samples were mounted and then sputter-coated with 5 nm Pt/Pd. Samples were then imaged on the Zeiss Gemini 360 SEM using the SE2 detector and accelerating voltage of 1.5 keV.

### PCR

The forward and reverse primers were prepared at 0.5 μM and probe oligo at 0.15 μM, respectively. Then, the PCR master mix (Prime Time Gene Expression Master Mix (2X), IDTDNA) was added to a 1X final concentration for PCR. The thermal cycle shown in Table 1 was used for running PCR (BioRad CFX 96). TE (pH 8.0) buffer was used for dilutions when necessary. The nucleic acid sequences of primers and probes are listed in Table 2.

### In vivo transduction study

100 μL of the RBC/AAV formulation containing 1e + 11 vg of AAV6-CMV-GFP was administered via the tail vein of C57BL/6 mice using a syringe with a 29 G needle. 450 ng of VEGF was included in all VEGF-containing formulations.

For carotid artery injections, carotid artery-catheterized C57BL/6 mice (Surgery code: CARART-CD; catheter inserted into the left carotid artery and advanced toward the aortic arch), aged 10–11 weeks and weighing 20–25 g, were purchased from Charles River Laboratories (Wilmington, MA). 1.51e + 10 vg of AAV9-CMV-dsRED was prepared in 100 μL of PBS and was injected via the catheter as free AAVs or loaded in RBC/AAV. Following the manufacturer's instructions, each injection was done into the Vascular Access Button (Instech) using 1 mL syringes via PinPorts (Instech; PNP3M) over 2 min to prevent clogging and pressure build-up at the peripheries.

### In vivo biodistribution study

1e + 11 vg of AAV6-CMV-Luc was IV injected via the tail vein. For the in situ drug loading condition, 20 μg TA and 6 μg $FeCl_3$ were co-injected. 1 h after administration, the mice were sacrificed by $CO_2$ euthanasia. Immediately after the euthanasia, mice were perfused with 20 mL of cold PBS, and the organs, including the brain, lung, heart, spleen, kidney, and liver, were collected. These were homogenized in TE buffer using a homogenizer (IKA T10 Basic ULTRA-TURRAX, NC). The viral genome was extracted from the homogenates using a DNA extraction kit (Thermo Scientific, K0721, GeneJet Genomic DNA Purification kit) and quantified using qPCR.

### Flow cytometry

3 weeks after RBC/AAV administration, mice were euthanized via $CO_2$ euthanasia. AAV6-CMV-GFP was used. The lungs were harvested after perfusion and placed in PBS containing 1% FBS. The lungs were cut into small pieces (<1 mm), then digested and processed using a lung dissociation kit (Miltenyi, 130-095-927) into single cells. Briefly, digestive enzymes were added to the samples and then incubated at 37 °C for 30 min. The samples were placed on top of pre-wet 70 μm strainers, mashed with plungers, and rinsed with 10 mL PBS. The cell suspensions

**Table 3 | List of antibodies for flow cytometric analysis of the lungs**

| Target | Fluorophore | Manufacturer | Catalog # | Clone | Dilution |
|---|---|---|---|---|---|
| CD326 | AF594 | Bioss | BS-4889R-A594 | unspecified | 1:80 |
| CD31 | BV711 | eBioscience | 407-0311-82 | 390 | 1:80 |
| CD45 | BUV395 | BD | 564616 | 104 | 1:80 |
| GFP | AF488 | BioLegend | 338007 | FM264G | 1:160 |
| CD16/32 | unconjugated | eBioscience | 14-0161-85 | 93 | 1:30 |

were collected in 50 mL tubes and centrifuged at 500 g for 5 min. The supernatant was removed, and the samples were washed twice with a cell staining buffer (BioLegend, 420201). If necessary, red blood cells were lysed with ACK buffer (Gibco, A1049201) before the washing step.

Following a typical flow cytometry sample preparation protocol, the resulting cells were stained with antibodies. Briefly, sample pellets were resuspended with 20 μL of 1/30 diluted anti-CD16/32 antibody (Invitrogen, 14-0161-85) for 10 min at 4 °C to block Fc receptors. Next, 20 μL of the antibody cocktail (list of antibodies shown in Table 3 for lung and brain samples, respectively) was added and incubated for 25 min at 4 °C. For the cocktail, each antibody was diluted by 1/40, which was then diluted to a final dilution of 1/80 once mixed with the sample solution containing CD16/32-coated cells. After the incubation, the samples were washed twice with PBS at 250 g for 5 min via centrifugation, followed by live/dead staining with LIVE/DEAD Fixable Blue (Invitrogen, L23105). Next, the samples were fixed and permeabilized using the BD perm/fix kit (BD, 554714) following the manufacturer's protocol for intracellular staining. Anti-GFP antibody was diluted to 1/160 for staining. After the staining, samples were washed twice with the perm/wash buffer, then finally resuspended in 200 μL of the staining buffer and stored at 4 °C until analysis.

The obtained flow cytometry data were analyzed using FlowJo software version 10.10.0. Gating strategy is described in Supplementary Methods and Supplementary Fig. 25.

### Binding study in whole blood

Binding of polystyrene beads on RBC in whole blood in a static condition was done by mixing 5 μL of fluorescent polystyrene beads (Fluorobrite®, cat# 17151; 200 nm diameter, $5.68 \times 10^{12}$ particles/mL), 10 μg TA and 3 μg Fe to a final volume of 10 μL of PBS (pH 7.4), followed by mixing with 10 μL of whole blood. The mixture was immediately diluted with 200 μL of PBS containing 5 w/v% BSA to quench further binding of the beads. The samples were washed twice with PBS. The fluorescence from the resulting samples was measured with a plate reader (BioTek Synergy H1).

For the binding study in the presence of flow, 77 μL of fluorescent polystyrene beads ($5.68 \times 10^{12}$ particles/mL) mixed with 100 μg TA and 30 μg Fe was injected into whole blood flowing in the tubing (0.02 inches inner diameter) at a 10 μL/s flow rate by a syringe pump, connected via a 25 gauge needle. The samples from the outlet were collected into PBS containing 4 w/v% BSA and were run immediately on a flow cytometer. Red blood cells were gated based on size, followed by gating on the PS bead-positive population.

For studying the binding of AAV to RBC in whole blood, Alexa Fluor 647-labeled AAV6-CMV-GFP was used. 8e + 9 vg of AAV was mixed with 16 μg TA and 4.8 μg Fe to a final volume of 8 μL, then mixed into 10 μL of whole blood. The samples were quenched with PBS with 5 w/v% BSA. The resulting samples were fixed with 4% PFA for 10 min at room temperature. Next, the RBC membrane was stained with a lipophilic dye, DiO, following the manufacturer's protocol. The resulting samples were run on a flow cytometer (Cytek Aurora).

### Statistics & reproducibility

All quantitative data are presented as mean ± SEM, unless otherwise stated. Statistical analyses were performed using GraphPad Prism 10 (GraphPad Software). Comparisons between two groups were analyzed using unpaired, two-sided Student's *t*-tests. Comparisons among more than two groups were analyzed using one-way or two-way ANOVA, followed by the indicated multiple comparisons test (Tukey's or Sidak's). Exact p-values are reported in each figure.

Sample sizes were chosen based on commonly used group sizes in the field and previous studies using RBC hitchhiking and AAV gene delivery; no statistical method was used to predetermine sample size. No data were excluded from the analyses. Animals were randomly assigned to experimental groups. Investigators were not blinded to allocation during outcome assessment, as treatment groups were apparent during experimental procedures. However, all outcome measures were obtained using standardized, quantitative assays (e.g., PCR, ELISA, flow cytometry), which minimized the potential for observer bias.

Representative images are shown from experiments that were independently repeated at least twice with similar results unless otherwise stated, as detailed in the corresponding figure legends.

### Reporting summary

Further information on research design is available in the Nature Portfolio Reporting Summary linked to this article.

## Data availability

Source data are provided with this paper. All data supporting the findings of this study are available within the manuscript, Supplementary Information, and Source Data file. Source data are provided with this paper.

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

## Acknowledgments

We thank the Harvard John A. Paulson School of Engineering and Applied Sciences Molecular and Cellular Biology Core (SEAS MCB) for infrastructure and support. SM and KF acknowledge support from the Rett Syndrome Research Trust. BS acknowledges support from Hitch Bio. MM and MEJ acknowledge support from the National Science Foundation Graduation Research Fellowship. A.D.C. acknowledges support from the Sara Borell grant (CD20/00054) and the fellowship for "International Researchers Mobility" (MV22/00079) from Carlos III Health Institute, Spain. We also thank Dana-Farber/Harvard Cancer Center in Boston, MA, for using the Rodent Histopathology Core, which provided histopathological interpretation. Dana-Farber/Harvard Cancer Center is supported in part by an NCI Cancer Center Support Grant # NIH 5 P30 CA06516.

## Author contributions

K.S.P., V.C.S., J.K., D.R., M.M., M.J., Y.G., M.G.B., J.-W.C., R.L., M.E.J., M.E., and A.D.S.C. performed experiments. J.K., K.D.F., S.M., and B.S. provided conceptual advice and analysis. K.S.P. and V.C. analyzed the results. K.S.P. and S.M. prepared illustrations and wrote the manuscript draft. D.L.C., J.A.N.M.H., and K.D.F. provided resources, technical expertise, and conceptual advice. K.S.P., J.K., S.M., and B.S. contributed to conceptual development and edited the manuscript. All authors contributed to data interpretation and provided feedback on the manuscript. All authors approved the final version of the manuscript.

## Competing interests

SM holds equity in Hitch Bio and serves on its Board of Directors. CP and SM are inventors on US Provisional Patent Application 63/755,667, filed by Harvard University and currently pending, which covers aspects of the research described in this manuscript. SM is also an inventor on PCT Patent Application PCT/US21/34132, filed by Harvard University and currently pending, which covers related aspects of the work. The remaining authors declare no competing interests.
