## [Transparent Peer Review file · Nature Communications]

Multiscale Red Blood Cell Hitchhiking for Targeted Deep Tissue Gene Delivery in Lungs

Corresponding Author: Professor Samir Mitragotri

Version 0:

Reviewer comments:

Reviewer #1

(Remarks to the Author)

This manuscript presents a novel strategy — MARVEL (Multiscale Approach using RBC-mediated hitchhiking and Vascular Endothelium Leakage) — to improve AAV-based gene delivery by leveraging red blood cell (RBC) hitchhiking in combination with VEGF-induced vascular permeabilization. The work is timely, well-conceived, and addresses a critical bottleneck in the field of systemic gene delivery: achieving tissue-specific, deep penetration in organs like the lungs. The methodology is innovative, and the manuscript is overall clearly written with robust supporting data. There are some conceptual clarifications, experimental controls, and mechanistic insights that should be addressed to improve the strength and generalizability of the findings before publication.

1. While the manuscript positions MARVEL as orthogonal to tropism-engineered AAVs, comparative text and/or data to such vectors (e.g., AAV9) would be valuable to contextualize its performance.
2. The authors demonstrate that both positively charged VEGF and negatively charged FITC-BSA can be loaded onto RBCs via the MARVEL formulation. What is the underlying mechanism? A brief discussion or citation would help explain the versatility of the platform and its potential for loading diverse therapeutic cargoes with opposite electrostatic properties.
3. The in situ approach offers excellent translational potential. More details on the mechanism of AAV-VEGF-RBC interactions in whole blood could be added. For example, deeper characterization of binding kinetics, loading efficiency over time, and competition with plasma proteins could strengthen the claims.
4. The term “Multiscale” could be more clearly defined early on—especially what scales are being bridged (molecular, cellular, vascular, etc.).
5. The manuscript refers to the use of AAVs throughout but does not consistently specify the serotype or construct in the main text or Materials and methods section.

Reviewer #2

(Remarks to the Author)

Samir Mitragotri, Bijay Singh and co-workers have developed a new approach to delivery AAVs and other cargos to difficult to reach target tissues and cells. Since this is an important limitation for the efficient development of gene therapies, including mRNA delivery, any further progress would be relevant.

This study takes advantage from previous work from the same lab and other laboratories that have loaded drug on red blood cells (RBC), called RBC hitchhiking, and providing evidence that this approach increases the accumulation of nanoparticles in various organs, including the lungs, brain, and kidneys. To overcome the endothelial barrier and increase further the delivery of the selected cargos by leading to deep tissue penetration of AAVs. To reach this goal the RBC is armed not only with the AAVs but also with the vascular endothelial growth factor (VEGF) known to improve endothelial permeability through the VEGF receptor-2.

The loading of AAVs and VEGF are obtained by concurrent adsorption on the RBC by complexation with tannic acid (TA) and iron chloride (III) (FeCl₃) to induce metal-phenolic network formation. From the technical point of view chemical cross-

linkers, including tannic acid, chromium chloride and iron chloride were employed in the 1950s for coupling of antigens and antibodies to RBCs to produce reagents suitable to study antibody mediated agglutination. Certainly the conditions used were different but these should be mentioned. In fact, over the years these approaches were surpassed by more specific approaches that avoid RBC membranes alterations.

The paper describe in the supplementary material the investigation that best permits to avoid hemolysis of the RBC and reduce aggregation of the complexes. This is appreciable but not entirely convincing.

The SEM images of TA/Fe/AAV complex on the RBC suggest a large and extensive modification of the RBC and this aspect should be investigate in more details. Profounds modifications as reported here have important effects on the rheology of RBC and on the immunogenicity of the same. The data reported from in vivo experiments are obtained after 1 hour from administration and apparently at this time the majority of modified RBC have been already removed from circulation. This is far from a physiological half-life of murine RBC.

Further data should support also the claims related to the loading of other model therapeutic cargoes, suggested in the discussion section including proteins, polymers, and LNPs, onto the RBC surface in whole blood.

Minor points:

A note of caution about the safety of the AAVs vectors should be considered since today Elevidys (Sarepta) caused the death of a patient. The therapy also in this case is delivered via an infusion into the bloodstream, using a modified AAV known as AAVrh74.

More recently other approaches have used RBC extracellular vesicles to deliver nucleic acids and this should likely be noted in the general discussion (<https://doi.org/10.1038/s41598-024-65623-y>)

Reviewer #3

(Remarks to the Author)

RBC hitchhiking is widely published by the co-author. This paper adds an innovative aspect in the form of MARVEL, where it can be used to deliver AAV in situ and leverage VEGF to overcome the endothelial barrier.

This approach is said to be liver-targeting; However, it seems that there is a high level of off-target distribution to the liver and spleen (Fig 5C), even though the levels are higher with AAV only. A side by side comparison of lung, liver and spleen distribution of the MARVEL cohort seems similar, indicating that it is not solely lung-targeting. Following this, gene expression was demonstrated in the lung endothelial and epithelial cells. It would be important to note the gene expression in off-target organs (liver and spleen) at the same time point considering the off-target biodistribution and the mechanisms of AAV (episomes with sustained transgene expression).

Following this logic, the off-target effects of VEGF-loaded MARVEL were not demonstrated; only the lung data was shown in Figure 4C. Based on Fig 3H, RBC/AAV demonstrated off-target distribution to the liver and spleen, which are highly vascularized organs and would be impacted by off-target VEGF. Again, this would be concerning in a clinical setting due to the possibility of unwanted sustained transgene expression in off-target organs.

Also, is there a way to visualize the in situ AAV delivery to RBC and quantify the efficiency beyond downstream endpoint analysis?

There is a lack of demonstrated of therapeutic efficacy in a disease model.

Add sample sizes to figure legends

The discussion is lacking and should be expanded. What are the implications of long-term VEGF exposure in the target organ (lung) and off-target organs? Safety - only IL-6 and TNF were investigated; what about immune response as a whole? Would re-dosing be an option or needed and the safety of it considering the off-target effects? What are the long-term implications - this study seems to be focused on short-term effects only. What about translatability of human RBC binding?

Version 1:

Reviewer comments:

Reviewer #1

(Remarks to the Author)

I recommend to accept the article.

Reviewer #2

(Remarks to the Author)

Thank you for considering all points raised in the initial review. The Manuscript is now more clear and robust.

Reviewer #3

(Remarks to the Author)

My comments have been addressed. For Fig. R6, it would have been better to obtain a quantitative scoring for the histology,

e.g., non-neoplastic inflammation scoring, necrosis etc, in at least n=3. The authors provided n=2, and n=1 depending on the group. Further, complete blood count and Superchem serum analyses would have strengthened the toxicity section.

RESPONSE TO THE REVIEWERS

We appreciate the reviewers' thoughtful feedback on our manuscript. We addressed the comments to the best of our ability.

REVIEWER COMMENTS

Reviewer #1 (Remarks to the Author):

This manuscript presents a novel strategy — MARVEL (Multiscale Approach using RBC-mediated hitchhiking and Vascular Endothelium Leakage) — to improve AAV-based gene delivery by leveraging red blood cell (RBC) hitchhiking in combination with VEGF-induced vascular permeabilization. The work is timely, well-conceived, and addresses a critical bottleneck in the field of systemic gene delivery: achieving tissue-specific, deep penetration in organs like the lungs. The methodology is innovative, and the manuscript is overall clearly written with robust supporting data. There are some conceptual clarifications, experimental controls, and mechanistic insights that should be addressed to improve the strength and generalizability of the findings before publication.

1. While the manuscript positions MARVEL as orthogonal to tropism-engineered AAVs, comparative text and/or data to such vectors (e.g., AAV9) would be valuable to contextualize its performance.

A1. The degree of association of the payload with TA may vary depending on its physicochemical properties. However, we anticipate that it will generally follow the same trend for a given class of payloads, for example, AAVs. In this study, we used two AAVs (AAV6 and AAV9), and the overall behavior of the interaction of these vectors with RBC was comparable (**Figure R1**). In response to the reviewer's comment, we have added text in the revised manuscript (**Page 6, paragraph 2**) and as the figure below as **Fig. S3**.

Figure R1 Loading AAV6-CMV-Luc and AAV9-CMV-dsRED on RBCs. The loading condition described in the methods section was used. Data are consolidated from three independent experiments. An unpaired Student's t-test was used for statistical analysis.

2. The authors demonstrate that both positively charged VEGF and negatively charged FITC-

BSA can be loaded onto RBCs via the MARVEL formulation. What is the underlying mechanism? A brief discussion or citation would help explain the versatility of the platform and its potential for loading diverse therapeutic cargoes with opposite electrostatic properties.

A2. Polyphenols can interact with proteins through non-covalent interactions, including hydrogen bonding, hydrophobic and electrostatic interactions, and van der Waals forces, which gives them versatility in loading proteins with diverse physicochemical properties (1). We have included this perspective in the manuscript (**Page 12, paragraph 1**), highlighting the versatility of our platform in loading cargoes with diverse properties.

3. The in situ approach offers excellent translational potential. More details on the mechanism of AAV-VEGF-RBC interactions in whole blood could be added. For example, deeper characterization of binding kinetics, loading efficiency over time, and competition with plasma proteins could strengthen the claims.

A3. Unlike chemical reactions that require energy to break and form covalent bonds, polyphenols, such as tannic acids, usually interact through weaker, reversible forces, such as hydrogen bonding, and hydrophobic interaction, which don't require activation energy of the magnitude that chemical reactions do. These reactions typically occur on a much shorter time scale than chemical reactions.

In response to the reviewer's comment, we conducted a binding kinetics assay by mixing rhodamine-tagged mouse IgG with whole blood and incubating for varied periods (10, 20, 30, 45, 60, 120, 240, and 480 seconds). The binding was saturated within 10 seconds, indicating rapid binding between the IgG and RBC mediated by tannic acid (**Figure R2**). This aligns with our observation in the study demonstrated in Figure S12, where polystyrene beads bound to RBCs under flow condition of whole blood within seconds.

Figure R2 Loading of rhodamine-tagged mouse IgG on RBC in whole blood. 10 μ L of IgG and whole blood were mixed and incubated for a varied time, immediately after which the solution was mixed with 5% BSA to quench the binding, centrifuged at 1500 g for 1 min, followed by washing with PBS. The resulting pellets were resuspended in PBS for fluorescence measurement to quantify the loaded IgG.

We also investigated the effect of serum proteins on loading efficiency by varying the serum concentrations using PS beads as a model drug (**Figure R3**). The binding efficiency dropped by about 2-fold when 10% serum was added to the medium, indicating that serum proteins compete for binding. Further addition of serum did not further decrease binding. Notably, TA and Fe (MARVEL) significantly increased the binding of PS beads to RBC in all serum conditions (**Figure R3A**). The fold increase in binding achieved through MARVEL increased with increasing serum concentration, up to 10-fold at 30% serum concentration, remaining at that level even for 60% serum concentration (**Figure R3B**). We have included this perspective in the manuscript (**Page 9, paragraph 3**) and added Figure R3 to the supplementary information as **Figure S15**.

Figure R3. Competition binding assay in varied serum concentrations. RBCs at 10% hematocrit in varying serum concentrations were mixed with 4.37×10^{10} fluorescent polystyrene (PS) beads with or without TA and Fe. Immediately after mixing, excess volume of PBS (5% BSA) was added to quench the reaction. **(A)** Total number of loaded PS beads by serum percentage. **(B)** Fold increases in loading after TA and Fe addition.

4. The term “Multiscale” could be more clearly defined early on—especially what scales are being bridged (molecular, cellular, vascular, etc.).

A4. Arteries and veins typically have 0.5-2.5 cm thicknesses, while the capillary vessels are approximately 5-10 μm . The payloads hitchhiking on an RBC freely migrate through the larger blood vessels (arteries and veins) until they reach the capillaries, where the RBCs must squeeze through to pass. The initial contact between the RBC and the endothelial cells at the target tissue occurs here, during which the payload is released and transferred.

For the subsequent infiltration into the deeper regions of the tissue, the payload has to migrate either transcellularly through the endothelial cells or paracellularly between cellular junctions, which occur on sub- μm -to-nm scales. We achieved a junctional breach that enhanced AAV migration into the deeper tissue by including VEGF, a known vascular permeabilizer.

Therefore, MARVEL's multiscale drug delivery is achieved by bridging the μm and nm scales. Per the reviewer's suggestion, we have clarified this earlier in the manuscript (**Page 5, paragraph 1**).

5. The manuscript refers to the use of AAVs throughout but does not consistently specify the serotype or construct in the main text or the Materials and Methods section.

A5. We revised the main text and the Materials and Methods section to clarify the serotypes of AAVs used in our studies.

Reviewer #2 (Remarks to the Author):

Samir Mitragotri, Bijay Singh, and co-workers have developed a new approach to the delivery of AAVs and other cargos to difficult-to-reach target tissues and cells. Since this is an important limitation for the efficient development of gene therapies, including mRNA delivery, any further progress would be relevant. This study takes advantage of previous work from the same lab and other laboratories that have loaded drugs on red blood cells (RBC), called RBC hitchhiking, and provides evidence that this approach increases the accumulation of nanoparticles in various organs, including the lungs, brain, and kidneys. To overcome the endothelial barrier and increase further the delivery of the selected cargos by leading to deep tissue penetration of AAVs. To reach this goal, the RBC is armed not only with the AAVs but also with the vascular endothelial growth factor (VEGF) known to improve endothelial permeability through the VEGF receptor-2. The loading of AAVs and VEGF are obtained by concurrent adsorption on the RBC by complexation with tannic acid (TA) and iron chloride (III) (FeCl_3) to induce metal-phenolic network formation.

1. From the technical point of view chemical cross-linkers, including tannic acid, chromium chloride and iron chloride were employed in the 1950s for coupling of antigens and antibodies to RBCs to produce reagents suitable to study antibody mediated agglutination. Certainly the conditions used were different but these should be mentioned. In fact, over the years these approaches were surpassed by more specific approaches that avoid RBC membranes alterations.

The paper describe in the supplementary material the investigation that best permits to avoid hemolysis of the RBC and reduce aggregation of the complexes. This is appreciable but not entirely convincing.

A1. We have added a sentence describing the previous studies that used polyphenols (e.g., tannic acid) to load proteins onto RBC surfaces, with relevant references listed below (**Page 4, paragraph 2**) (2, 3). In response to the reviewer's comment, we have also performed an additional safety evaluation. This is discussed in the next point.

- S. V. Boyden, The adsorption of proteins on erythrocytes treated with tannic acid and subsequent hemagglutination by antiprotein sera. J Exp Med 93, 107-120 (1951).

- F. Herz, E. Kaplan, Effects of tannic acid on erythrocyte enzymes. *Nature* 217, 1258-1259 (1968).

2. The SEM images of TA/Fe/AAV complex on the RBC suggest a large and extensive modification of the RBC and this aspect should be investigated in more details. Profound modifications as reported here have important effects on the rheology of RBC and on the immunogenicity of the same. The data reported from in vivo experiments are obtained after 1 hour from administration and apparently at this time the majority of modified RBC have been already removed from circulation. This is far from a physiological half-life of murine RBC.

A2.

Figure R4. Viscosity of RBC at 10% hematocrit. Statistical analysis was done using a two-way ANOVA, followed by Sidak's multiple comparisons test (* $p < 0.05$ and **** $p < 0.0001$).

We investigated the rheology of RBC by measuring viscosity at 10% hematocrit, with and without TA and Fe coating (**Figure R4**). The TA/Fe coating on RBC had a minor effect on viscosity, leading to significant differences only at very low shear rates ($< 10 \text{ s}^{-1}$). There was no statistically significant difference in viscosity at physiological shear rates (typically $> 40\text{-}100 \text{ s}^{-1}$ in capillaries and $> 100\text{-}1000$ in arteries) (4, 5). We have included this in the text (**page 5, paragraph 1**) and Supplementary Figures as Fig. S2.

However, we agree with the reviewer that extensive modification to the RBC surface may affect their half-life, leading to immunogenicity or toxicity. Therefore, we conducted some additional experiments to examine the immunogenicity and toxicity.

Concerning the immunogenicity of the modified RBC, we measured the IgG responses after IV injection of either free AAV or RBC/AAV complex (**Figure R5**). We did not observe a significant difference in IgG levels between the two groups, demonstrating that the RBC modification does not exacerbate the immunogenicity of AAV.

Figure R5. Serum IgG titer of free AAV or RBC/AAV-injected mice. Two IV injections were given on days 0 and 28, followed by sera collection on day 56. ELISA against mouse total IgG was performed on serially diluted sera. EC50 (reciprocal serum dilution) was used to quantify the serum IgG titers. N=8 per group. An unpaired Student's t-test was used for statistical analysis.

We conducted a histology analysis on major organs, including the brain, lungs, heart, liver, kidneys, and spleen, through H&E staining to test potential toxicity induced by the system (**Figure R6**). Evaluation by a pathologist reported no observable toxicities.

Figure R6. Histology sections of major organs, harvested 30 days post-IV injection. For the in situ MARVEL formulation, organs were harvested on days 10 and 30 after IV injection to evaluate short and long-term toxicity. N=2 (and n=1 for MARVEL 10d).

We have included the IgG titer data in the text (page 7, paragraph 2) and Supplementary Figures as Fig. S8. Histology data have also been added to the text (page 11, paragraph 1) and Supplementary Figures as Fig. S21.

3. Further data should support also the claims related to the loading of other model therapeutic cargoes, suggested in the discussion section including proteins, polymers, and LNPs, onto the RBC surface in whole blood.

A3. Figures 5A and S18 show the polymer (polystyrene) and LNP loading on the RBC surface in whole blood. We have conducted a study confirming a model protein drug, mouse IgG, loading on RBC in whole blood (Figure R2).

Minor points:

A note of caution about the safety of the AAV vectors should be considered since today, Elevidys (Sarepta) caused the death of a patient. The therapy, also in this case, is delivered via

an infusion into the bloodstream using a modified AAV known as AAVrh74. More recently, other approaches have used RBC extracellular vesicles to deliver nucleic acids, and this should likely be noted in the general discussion (<https://doi.org/10.1038/s41598-024-65623-y>)

A. In response to the reviewers' comments, we have noted caution in the manuscript about the safety concerns of AAV vectors. We have also cited the paper suggested by the reviewer in the discussion section as an update to recent advances in nucleic acid delivery using RBC. These are indicated in the text (**page 12, paragraph 2**).

Reviewer #3 (Remarks to the Author):

RBC hitchhiking is widely published by the co-author. This paper adds an innovative aspect in the form of MARVEL, which can be used to deliver AAV in situ and leverage VEGF to overcome the endothelial barrier. This approach is said to be liver-targeting; However, it seems that there is a high level of off-target distribution to the liver and spleen (Fig 5C), even though the levels are higher with AAV only. A side-by-side comparison of lung, liver, and spleen distribution of the MARVEL cohort seems similar, indicating that it is not solely lung-targeting. Following this, gene expression was demonstrated in the lung endothelial and epithelial cells.

1. It would be important to note the gene expression in off-target organs (liver and spleen) at the same time point considering the off-target biodistribution and the mechanisms of AAV (episomes with sustained transgene expression).

A1. We have taken images of transgene expression from the spleen and liver 21 days after carotid arterial injection or 30 days after intravenous injection (**Figures R7 and R8**).

For the transgene expression in the liver and spleen after carotid arterial injection, mice injected with free AAV or RBC/AAV using AAV9-CMV-dsRED were compared (**Figure R7**). There was no statistical difference in transgene expression in both organs. We have included this in the text (**page 7, paragraph 2**) and Supplementary Figure as **Fig. S7**.

Regarding the transgene expression in the liver after intravenous injection, mice injected with free AAV or RBC/AAV with and without VEGF were compared using AAV6-CMV-GFP (**Figure R8**). There were statistically significant increases in liver transgene expression compared to the untreated control when VEGF was added to the formulation in both the free AAV and RBC/AAV. Interestingly, although the addition of VEGF led to increased transgene expression in the liver, RBC hitchhiking itself did not increase the transgene expression compared to soluble counterparts, i.e., soluble AAV with or without VEGF. This aligns with what is observed in Figure 3H from the biodistribution study, indicating that RBC hitchhiking does not contribute to significant off-target

gene expression. We have included this insight in the text (page 10, paragraph 3) and Supplementary Figures as Fig. S20.

Figure R7. Transgene expression in the liver and spleen 21 days after CA injection of 1.51×10^{10} vg AAV9-CMV-dsRED formulations. (A) Ex vivo images of the livers and spleens. (B) Average radiant efficiency of dsRED signals from the liver and spleen. N=4 per group. An unpaired Student's t-test was performed for statistical analyses.

Figure R8. Transgene expression in the liver 30 days after IV injection of AAV6.GFP formulations. (A) Ex vivo images of the livers. (B) Average radiance of GFP signals from the liver. N=5 per group. One-way ANOVA, followed by Tukey's multiple comparisons test, was performed for (B).

2. Following this logic, the off-target effects of VEGF-loaded MARVEL were not demonstrated; only the lung data was shown in Figure 4C. Based on Fig 3H, RBC/AAV demonstrated off-target distribution to the liver and spleen, which are highly vascularized organs and would be impacted by off-target VEGF. Again, this would be concerning in a clinical setting due to the possibility of unwanted sustained transgene expression in off-target organs.

A2: As the reviewer mentioned, there is a possibility that there could be off-target accumulation of VEGF in the liver and spleen, based on Figure 3H. However, based on Figure 3G, the amounts of off-target accumulation are similar to or lower than those of soluble controls. Nonetheless, the prolonged persistence of VEGF in highly vascularized organs may induce toxicity, which is indeed of potential concern in clinical settings. We, therefore, added histology images of major organs that were harvested on days 10 and 30 after IV injection of formulations tested in Figure 5E to evaluate the toxicities. We have added the relevant data in **Fig. S21**.

3. Also, is there a way to visualize the in situ AAV delivery to RBC and quantify the efficiency beyond downstream endpoint analysis?

A3. When compared to the number of endogenous RBCs within the blood, the amount of cargo to be delivered via in situ hitchhiking is significantly less, which makes it challenging to detect the cargo that has been successfully loaded on the RBC surface en route to the target capillary bed in the lungs. In mice, it takes approximately 10-20 seconds for the entire blood to circulate through the body. The time for the formulation to reach the lungs after administration would be even shorter, further complicating the visualization of in situ hitchhiking other than the endpoint analysis.

As an alternative, we used an in vitro loading study using whole blood to evaluate the approximate time it takes for a drug to interact with the RBC in the presence of flow. Based on the study demonstrated in Figures S11A and B, the binding occurred relatively quickly (< 2 sec).

In addition, using a similar condition used for loading fluorescent polystyrene (PS) beads on RBCs in whole blood, described in Figure 5A, we mixed whole blood with a mixture of PS beads, TA, and Fe, and ran flow cytometry. Next, we gated on the bead+ population and imaged them using FACS Discover S8 (BD) to visualize the binding of PS beads on RBCs (**Figure R9**). We added this description in text (**page 10, paragraph 1**) and the relevant data in **Fig. S17**.

Figure R9. Real-time imaging of model drug (PS beads) bound to RBC in whole blood. (A) Flow cytometry plots of neat whole mouse blood (upper) or pulse-mixed with PS beads, TA, and Fe (lower). **(B)** Bead-positive population was gated and imaged in real-time using BD FACS Discover S8.

4. There is a lack of demonstrated therapeutic efficacy in a disease model.

A4. We agree that evaluating therapeutic efficacy in a relevant disease model is an important next step. However, the current study is focused on validating the concept of VEGF-induced deep tissue transduction. We plan to explore its therapeutic applicability in future studies.

5. Add sample sizes to figure legends

A5. We added sample sizes in the figure legends.

6. The discussion is lacking and should be expanded. What are the implications of long-term VEGF exposure in the target organ (lung) and off-target organs? Safety - only IL-6 and TNF were investigated; what about the immune response as a whole? Would re-dosing be an option or needed and the safety of it considering the off-target effects? What are the long-term implications - This study seems to focus only on short-term effects. What about the translatability of human RBC binding?

A6. To address these questions, we performed histological analyses on both target and off-target organs, including the lungs, brain, heart, liver, kidneys, and spleen, on days 10 and 30 following intravenous administration of the in situ hitchhiking formulation to evaluate toxicities (**Figure R6**). In

parallel, sera from mice injected with either free AAV or RBC/AAV formulations were collected to assess systemic immune responses via serum IgG levels (**Figure R5**). No significant differences were observed between the two groups, suggesting that RBC hitchhiking does not exacerbate AAV-induced immunogenicity. Furthermore, the enhanced and sustained transduction achieved via RBC hitchhiking may reduce the need for frequent dosing or even enable effective single-dose regimens, which could be advantageous in clinical settings. Finally, the applicability of RBC hitchhiking has been demonstrated in species beyond mice using model drugs, supporting its translational potential for AAV-mediated gene delivery (7). We added these points in the Discussion section (**page 12, paragraph 2**).

References

1. H. Xue *et al.*, Research progress on the interaction of the polyphenol–protein–polysaccharide ternary systems. *Chemical and Biological Technologies in Agriculture* **11**, 95 (2024).
2. S. V. Boyden, The adsorption of proteins on erythrocytes treated with tannic acid and subsequent hemagglutination by antiprotein sera. *J Exp Med* **93**, 107-120 (1951).
3. F. Herz, E. Kaplan, Effects of tannic acid on erythrocyte enzymes. *Nature* **217**, 1258-1259 (1968).
4. H. N. Mayrovitz, Skin capillary metrics and hemodynamics in the hairless mouse. *Microvasc Res* **43**, 46-59 (1992).
5. M. A. Panteleev *et al.*, Wall shear rates in human and mouse arteries: Standardization of hemodynamics for in vitro blood flow assays: Communication from the ISTH SSC subcommittee on biorheology. *J Thromb Haemost* **19**, 588-595 (2021).
6. Z. Zhao *et al.*, Red Blood Cell Anchoring Enables Targeted Transduction and Re-Administration of AAV-Mediated Gene Therapy. *Adv Sci (Weinh)* **9**, e2201293 (2022).
7. J. S. Brenner *et al.*, Red blood cell-hitchhiking boosts delivery of nanocarriers to chosen organs by orders of magnitude. *Nat Commun* **9**, 2684 (2018).

We appreciate the reviewers for their comments. We have addressed the points raised in the previous review, as outlined below.

Reviewer #3 (Remarks to the Author)

My comments have been addressed. For Fig. R6, it would have been better to obtain a quantitative scoring for the histology, e.g., non-neoplastic inflammation scoring, necrosis etc, in at least n=3. The authors provided n=2, and n=1 depending on the group. Further, complete blood count and Superchem serum analyses would have strengthened the toxicity section.

Response to the Reviewer #3

Thank you for this comment. For data shown in Fig. S21, please note that the experiments (saline, VEGF+AAV, MARVEL) were performed with N=5-7. Two animals were randomly chosen from these experiments for histological evaluation. This is now clarified in the revised manuscript (Revised **Fig. S21**).

In response to the reviewer's comment, we have quantified the histological scores, and these are included in the revised manuscript (**New Fig. S21B**).

In response to the reviewer's comment, we have performed an additional evaluation of serum chemistry and hematology after IV treatment of MARVEL at two time points (24 hours and 1 week). This is added as **New Fig. S22**. We've also added the description in the main text (**Page 11, paragraph 1**).